# Isolation, identification, and whole-genome sequencing of *Streptomyces rochei* FE-3-1 against *Pyricularia oryzae*

Dongxia Du[1,2], Zhuo Yi[3]*, Shiping Shan[1,2]*, Shuaishuai Gao[1], Mengyuan Yu[1], Bin Wang[1,2]

1 Hunan Institute of Microbiology, Changsha, Hunan, China, 2 Hunan Engineering and Technology Research Center of Agricultural Microbiology Application, Changsha, Hunan, China, 3 Yiyang Open University, Yiyang, Hunan, China

* yizhuo_2008@126.com (ZY); ssp312@hotmail.com (SS)

## Abstract

*Streptomyces* are significant producers of antimicrobial secondary metabolites. In this study, a *Streptomyces rochei* FE-3–1was isolated from the rhizosphere of rice plants, and its metabolites exhibited potent antagonistic activity against plant pathogen *Pyricularia oryzae*. However, the genome sequence of the strain has not been reported to date. Whole genome sequencing and genome mining were conducted to comprehensively characterize the strain's main features. The results showed that the total size of the genome is 8,247,561 bp with 72.51% G + C content. Among a total of 7158 genes, 169 predicted RNA genes were identified including 67 transfer RNA (tRNA) genes, 18 ribosomal RNA (rRNA) genes and 84 small RNA (sRNA) genes, as well as 14 genomic islands were predicted. A total of 31 biosynthetic gene clusters were detected within the genome of *Streptomyces rochei* FE-3–1, and at least four of these gene clusters are associated with known potent antimicrobials. These findings provide a solid theoretical foundation for utilizing strain FE-3–1 in developing biofertilizers or biopesticides within the field of biotechnology.

## 1. Introduction

With the increasingly extensive and frequent use of antibiotics, the overuse of antibiotics has become a serious global health problem [1]. Subsequently, due to the increasingly serious problem of antimicrobial resistance, and the research and development of novel antibiotics have been becoming more urgent [2].

 *Streptomyces* can survive in different habitats such as soil, plant bodies, volcano, desert, animal, limestone, sea water, and fresh water, etc [3]. *Streptomyces* strains have strong metabolic capacity and can better adapt to changing environments, and represent the biggest genus in the phylum of *Actinobacteria* [4]. It is a class of Gram-positive, high G + C content, and spore-forming aerobic *actinomycetes* with highly branched basal

**Editor:** Chetan Keswani, Southern Federal University Academy of Biology and Biotechnology named after D I Ivanovsky: Uznyj federal'nyj universitet Akademia biologii i biotehnologii im D I Ivanovskogo, RUSSIAN FEDERATION

**Data availability statement:** The data were uploaded to the NCBI GenBank under the accession numbers: BioProject (PRJNA1098797), BioSample(SAMN40923439), SRA (SRR28714279).

**Funding:** This study was supported by Agricultural Science and Technology Innovation Fund Project of Hunan Province in the form of a grant awarded to Shiping Shan (2025CX83) and the Project of Special Fund Project of the Department of Agriculture and Rural Affairs of Hunan Province in the form of a grant awarded to Shiping Shan (2025 Xiangcaiyu 0001). The specific roles of this author are articulated in the 'author contributions' section. The funders had no role in study design, data collection and analysis, decision to publish, or preparation of the manuscript.

**Competing interests:** The authors declare that they have no competing interests.

**Abbreviations:** KEGG, Kyoto Encyclopedia of Genes and Genomes; NCBI, National Center for Biotechnology Information; COG, Cluster of Orthologous Groups of Proteins; GO, Gene Ontology; rRNA, ribosomal RNA; tRNA, transfer RNA; sRNA, small RNA.

mycelia and gas mycelia, which is capable of metabolizing and synthesizing abundant secondary metabolites [5]. Several natural products have been derived from *actinomycetes* [6]. *Streptomyces* are the most significant producers of antibiotics among bacterial species. They also play a crucial role as agents for growth promotion and biocontrol, with their application in agriculture becoming increasingly prevalent [7,8].

While the genome of *Streptomyces* contains abundant gene clusters for secondary metabolites, its structure is complex and often experiences partial gene silencing, making it challenging to discover new active substances [9]. The analysis of genomes has revealed that *Streptomyces* species can harbor a range of 25–70 biosynthetic gene clusters [10]. Whole genome sequencing and genome mining can accurately predict the number and type of secondary metabolic gene clusters, the size, location, function and skeleton structure of coding genes, as well as metabolic regulation mode, which is conducive to the rapid mining of new metabolites [11]. Currently has 1290 *Streptomyces* genome published HTML (http://www.bacterio.net/streptomyces).

Numerous bioactive molecules have previously been isolated from *Streptomyces* species, exhibiting reported activities of secondary metabolites such as antifungal, antibacterial, anti-inflammatory, antitumor, and immunosuppressive effects [6]. Streptomycin and actinomycin were the first antibacterial compounds to be isolated from *Streptomyces* [12]. Currently, a variety of known antibiotics including streptomycin, tetracycline, erythromycin, chloramphenicol, lincomycin, kanamycin, clindamycin are derived from *Streptomyces* [13].

*Streptomyces* are the largest bacterial genus among the plant-associated *actinomycetes* isolated so far [14], and the roles of *Streptomyces* in the biological control of plant pathogens are also significant. Some metabolites of *Streptomyces* are active plant growth promotion [15]. Some *Streptomyces* can also produce enzymes that have extracellular cell wall degrading activity, thereby improving plant disease resistance [16,17].

Based on different *Streptomyces* strains, many commercial biocides, such as Actinovate (*Streptomyces lydicus*), Rhizovit (*Streptomyces* sp. DSMZ12424), and Mycostop (*Streptomyces griseoviridis*), have been developed for agricultural purposes [18–20]. Therefore, *Streptomyces* are important resources for agricultural biocides or biofertilizers.

In order to further understand the biological function and secondary metabolic potential of *Streptomyces rochei* FE-3–1, a genome-wide assay was performed and the complete genome sequence of the strain FE-3–1 was obtained. Gene prediction, functional annotation, secondary metabolic synthesis and comparative genome analysis were studied. This study provides a theoretical basis for further exploring its biocontrol potential and developing secondary metabolites that can be used for agricultural biological control.

## 2. Materials and methods

### 2.1. Isolation of *Streptomyces* strains

Gradient dilution separation was employed to isolate *Streptomyces* strains from rice rhizosphere soil samples [21]. Specifically, l g of soil sample was added into a flask containing 99 mL of sterile water, diluted at l:100 ratio, and shaken at 160 rpm for 30 min.

After standing, the supernatants were diluted with sterile water into $10^{-3}$, $10^{-4}$, and $10^{-5}$ fold dilutions. Gause No.l agar medium containing antifungal $K_2Cr_2O_7$ 75 µg/mL was spread with 200 µL of each soil dilution and incubated at 28°C for 5~7 days. The isolated single colonies were purified, cultured for 5 days, and transferred to Gause No.l agar medium for preservation.

## 2.2. Determination of anti-*Pyricularia oryzae* activity

The pathogenic fungi *Pyricularia oryzae* was inoculated onto PDA medium, and the fungal cake was prepared using a 6 mm hole punch and then inoculated at the center of the PDA plate. The strain FE-3–1 was streak-inoculated on two opposite edges from the center of the plate. The fungal cake of *Pyricularia oryzae* was separately inoculated as a control. Three replicates were conducted for each treatment. The colony radius (from the edge of the cake to the edge of pathogenic fungi) was determined by the cross-crossing method using vernier calipers, and inhibition rate was calculated. Relative inhibition rate (%) = (control pathogen colony radius – pathogen colony radius on the plate inoculated with antagonistic strain FE-3–1)/control pathogen colony radius ×100%.

## 2.3. Identification of the strain FE-3–1

The strain FE-3–1 was inoculated and purified in Gause No.1 agar medium, followed by cultivation at 28°C for 5~7 days to observe colony characteristics and color. Spore morphology and mycelium were also examined via optical microscopy. Total DNA was extracted from the purified FE-3–1 strain using QIAamp DNA Mini Kit (Qiagen, CA, USA) according to the manufacturer's protocol, and the 16S rDNA gene sequence was amplified via PCR with universal primers 27F and 1492R. The resulting DNA sequences were aligned using ClustalW, and a phylogenetic tree was constructed using MEGA 6.0 [22].

## 2.4. Genome assembly, scaffolding, and annotation

The whole genome sequence of the strain FE-3–1 was obtained through Pacbio and Illumina Hiseq×10 platforms with approximately 100-fold coverage in both platforms. Genomic DNA was randomly fragmented using Covaris or Bioruptor method, followed by sequencing adaptor ligation according to the manufacturer's instructions for Illumina Paired-End sequencing library preparation. The prepared libraries and genome were sequenced. The resulting reads were de novo assembled via SOAPdenovo v1.05. The genome was annotated by the NCBI Pro-karyotic Genome Annotation Pipeline, and genes were identified by GeneMarkS⁺.

## 2.5. Screening genes related to beneficial traits

Biosynthetic gene clusters responsible for synthesizing active secondary metabolites were predicted in the genome of the strain FE-3–1 using antiSMASH bacterial version 7.0 (http://antismash.secondarymetabolites.org/). Additionally, Prokka-annotated genes were screened for plant growth promotion traits, such as phosphorus solubilization, potassium solubilization, nitrogen assimilation, siderophore production, plant hormone production as well as biocatalysts including protease, lipase, chitinase, and catalase production. Furthermore, heavy metal resistance genes were also manually screened.

## 2.6. Gene functional category

Functional enrichment analysis classified core gene families into different biological functions based on COG/GO/KEGG databases. The numbers of corresponding proteins were computed for each term of COG/GO/KEGG, and the resulting results were visualized by GraphPad Prism 7.0.

## 2.7. Pan-genome and comparative genomic analysis

A pan-genome analysis involving 9 *Streptomyces*-related species' genomes was manipulated using Roary Pan-genome Pipeline, and regression analysis of core gene cluster curves was conducted by weighted least square regression.

### 2.8. Statistical analysis

Means and standard errors of the data were calculated using Excel 2010 (Microsoft Corporation, USA). All data collected were statistically analyzed according to Duncan's multiple range test (P = 0.05).

## 3. Results

### 3.1. Isolation and identification of FE-3–1

The colonies of the strain FE-3–1 are nearly round, dry, with protrusions in the middle, and opaque with a regular slick edge on Gause No.1 agar plates (Fig 1A). The spore mass is gray and the basal mycelium is orange. The strain FE-3–1 is Gram-positive and aerobic, with vigorously growing aerobic hyphae that have more branches (Fig 1B). The basic characteristics and classification of the strain FE-3–1 are shown in S1 Table in S1 File. Based on the 16S rDNA gene sequences, the phylogenetic tree of the strain FE-3–1 was constructed using neighbor-joining method, and phylogenetic analysis placed the strain FE-3–1 as most closely related to *Streptomyces rochei* NRRL B-2410 (Fig 2).

### 3.2. Anti-*Pyricularia oryzae* activity analysis

The strain FE-3–1 exhibited inhibitory effect on plant pathogen *Pyricularia oryzae* (Fig 3), indicating its potential as a source of anti-*Pyricularia oryzae* compound. The results demonstrated that the colony radius of pathogenic fungi *Pyricularia oryzae* on the plate inoculated with antagonistic strain FE-3–1 was 8.00 ± 0.035 mm, whereas that of control pathogen fungi was 30.00 ± 0.046 mm, resulting in an inhibition rate of 73.33 ± 0.079%.

### 3.3. Genome sequencing, annotation and features

The strain was selected for sequencing primarily due to its anti-*Pyricularia oryzae* activity. Genome was sequenced by Shanghai Majorbio Bio-pharm Technology Co., Ltd (Shanghai, China). The project information is summarized in S2 Table in S1 File. A standard shotgun library generated 341255 reads with an average length of 4546.05 bp. The total size of the genome is 8,247,561 bp with a G + C content of 72.51% (Fig 4). The genome properties and statistics are summarized in Table 1. Among the total of 7158 genes, there were 169 predicted RNA genes including 67 transfer RNA (tRNA) genes, 18 ribosomal RNA (rRNA) genes and 84 small RNA (sRNA) genes, as well as 14 genomic islands (S3 Table in S1 File). Functional annotation of genes from the strain FE-3–1 was performed using GO (Fig 5), COG (Fig 6) and KEGG (Fig 7) databases which revealed active transcription, metabolism, signal transduction, secondary metabolites biosynthesis and

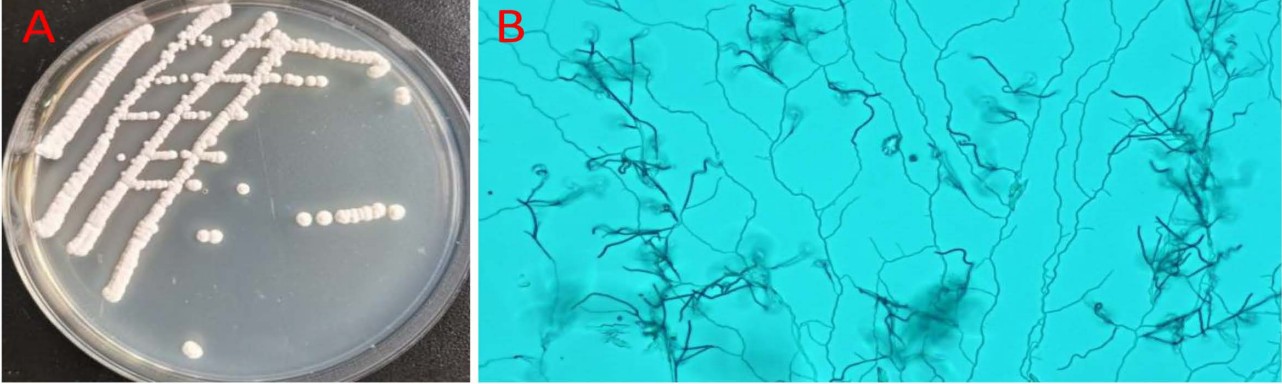

**Fig 1. The colony morphology of *Streptomyces rochei* FE-3-1 (A).** The morphology of hyphae of *Streptomyces rochei* FE-3-1 observed under 100-fold optical microscopy (B).

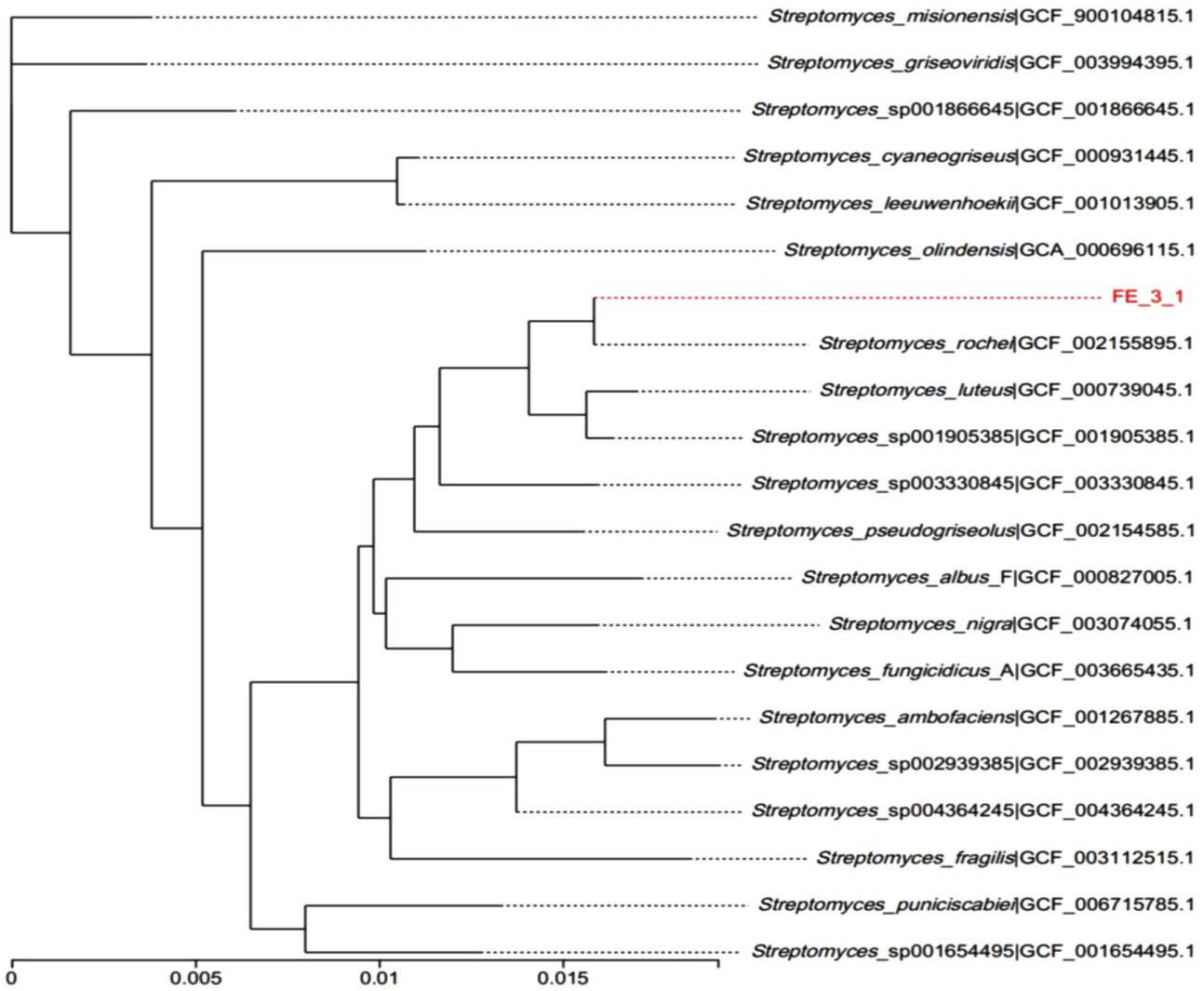

**Fig 2. The phylogenetic tree of *Streptomyces rochei* FE-3-1 based on the 16S rDNA gene sequence.**

membrane transport activities (Fig 5). Additionally, 5571 (77.83%) genes are distributed into COG functional categories (Fig 6). The 2846 annotated gene products are involved in 254 pathways, including metabolism (143 pathways), cellular processes (13 pathways), genetic information processing (16 pathways), organismal systems (30 pathways), human diseases (30 pathways), and environmental information processing (13 pathways).

### 3.4. Identification of heavy metal resistance genes

According to the results of genomic annotation, the strain FE-3–1 harbors multiple putative functional proteins associated with different heavy metals (Cadmium, Arsenic and Copper) or metal (Zinc), and putative functions including transporters, P-type ATPase and resistance proteins, and so on (Table 2). The strain exhibits resistance to a variety of heavy metals or metal, enabling it to effectively mitigate the impact of heavy metal or metal pollution, particularly in environments characterized by combined cadmium and arsenic contamination.

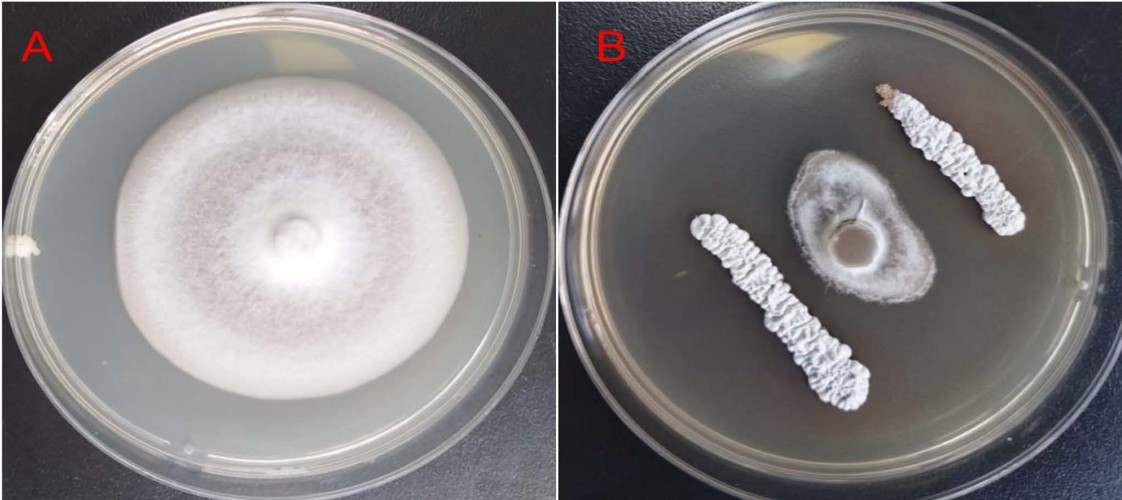

**Fig 3. The inhibitory effect of strain FE-3-1 on *Pyricularia oryzae*.**

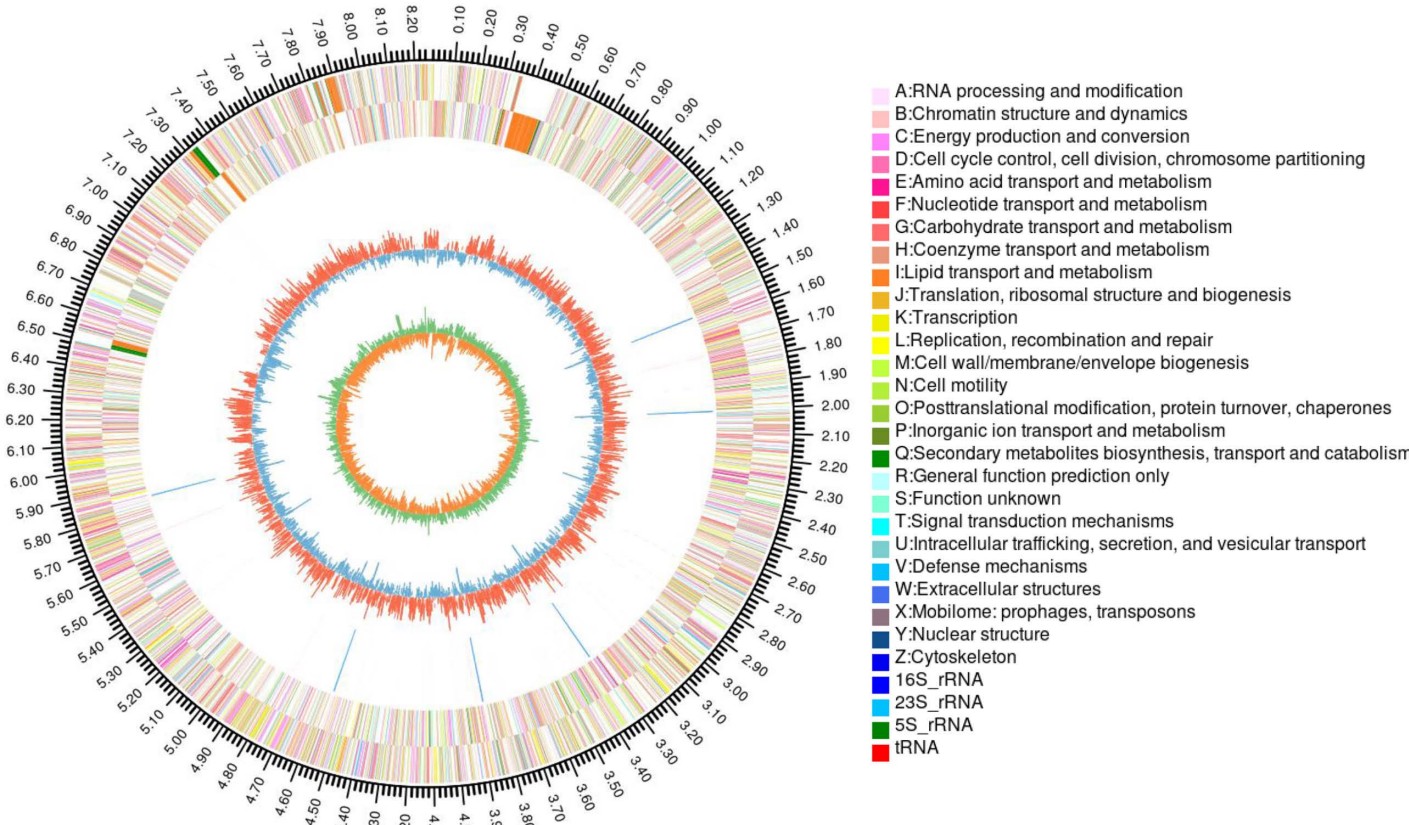

A:RNA processing and modification
B:Chromatin structure and dynamics
C:Energy production and conversion
D:Cell cycle control, cell division, chromosome partitioning
E:Amino acid transport and metabolism
F:Nucleotide transport and metabolism
G:Carbohydrate transport and metabolism
H:Coenzyme transport and metabolism
I:Lipid transport and metabolism
J:Translation, ribosomal structure and biogenesis
K:Transcription
L:Replication, recombination and repair
M:Cell wall/membrane/envelope biogenesis
N:Cell motility
O:Posttranslational modification, protein turnover, chaperones
P:Inorganic ion transport and metabolism
Q:Secondary metabolites biosynthesis, transport and catabolism
R:General function prediction only
S:Function unknown
T:Signal transduction mechanisms
U:Intracellular trafficking, secretion, and vesicular transport
V:Defense mechanisms
W:Extracellular structures
X:Mobilome: prophages, transposons
Y:Nuclear structure
Z:Cytoskeleton
16S_rRNA
23S_rRNA
5S_rRNA
tRNA

**Fig 4. A graphical circular map of *Streptomyces rochei* FE-3-1.** From outside to center, the outermost ring shows the marker of the genome size; rings 2, 3 show protein-coding genes colored by COG categories on forward/reverse strand; rings 4 shows rRNA and tRNA; ring 5 shows G + C % content plot; the innermost circle is the GC-Skew.

**Table 1. The genome properties and statistics of *Streptomyces rochei* FE-3-1.**

| Attributes | Values |
|---|---|
| Genome size (bp) | 8247561 |
| CDS No. | 7158 |
| G+C Content(%) | 72.51 |
| tRNA No. | 67 |
| Type of tRNAs No. | 20 |
| rRNA No. | 18 |
| Gene No. | 7158 |
| Gene total length(bp) | 7280481 |
| Gene average length(bp) | 1017.11 |
| Gene density(kb) | 0.87 |
| GC content in gene region(%) | 72.74 |
| Gene Len/Genome(%) | 88.27 |
| Intergenetic region length(bp) | 967080 |
| GC content in intergenetic region(%) | 70.81 |
| Intergenetic length/Genome(%) | 11.73 |
| Total reads num | 341255 |
| Average length of reads(bp) | 4546.05 |
| CRISPR-Cas No. | 17 |
| Total lengthof tandem repeat(bp) | 70328 |
| Tandem repeat/Genome(%) | 0.97 |
| Genes No. of Cellular Component | 1354 |
| Genes No. of Molecular Function | 2993 |
| Genes No. of Biological Process | 1460 |
| Genes assigned to COGs | 5571 |
| Genes with Pfam domains | 5889 |

## 3.5. Genes related to plant growth promotion traits

According to Prokka annotations, the genome of *Streptomyces rochei* FE-3–1 harbors numerous genes associated with promoting plant growth, including those involved in phosphate solubilization, potassium solubilization, nitrogen assimilation and siderophore production (Table 3). Additionally, the presence of genes responsible for chitinase production and related lyases, such as protease and lipase, suggests potent antifungal properties, and antifungal activity of the strain has been confirmed in Fig 1. Furthermore, the peroxidase/catalase-related genes were also detected within the genome of *Streptomyces rochei* FE-3–1 and the production of peroxidase and catalase by strain FE-3–1 may be an important mechanism of plant resistance to oxidative stress.

## 3.6. Prediction of secondary metabolites-related gene clusters

A total of 31 biosynthetic gene clusters were identified within the genome of *Streptomyces rochei* FE-3–1 via antiSMASH bacterial version 7.0 (Table 4). Four of these biosynthetic gene clusters exhibited high similarity to known metabolites with antifungal and antibacterial properties such as candicidin, streptothricin, lipopeptide, and albaflavenone (Fig 8). Some gene clusters were also related to active metabolites such as isorenieratene, desferrioxamine, hopene, ectoine, sapB, fluostatins, 7-prenylisatin, melanin, borrelidin and geosmin. Some gene clusters showed limited or no similarity to known metabolites. These types of gene clusters included terpene, indole, ectoine, melanin, arylpolyene, T2PKS, NRPS, lanthipeptides, and siderophores.

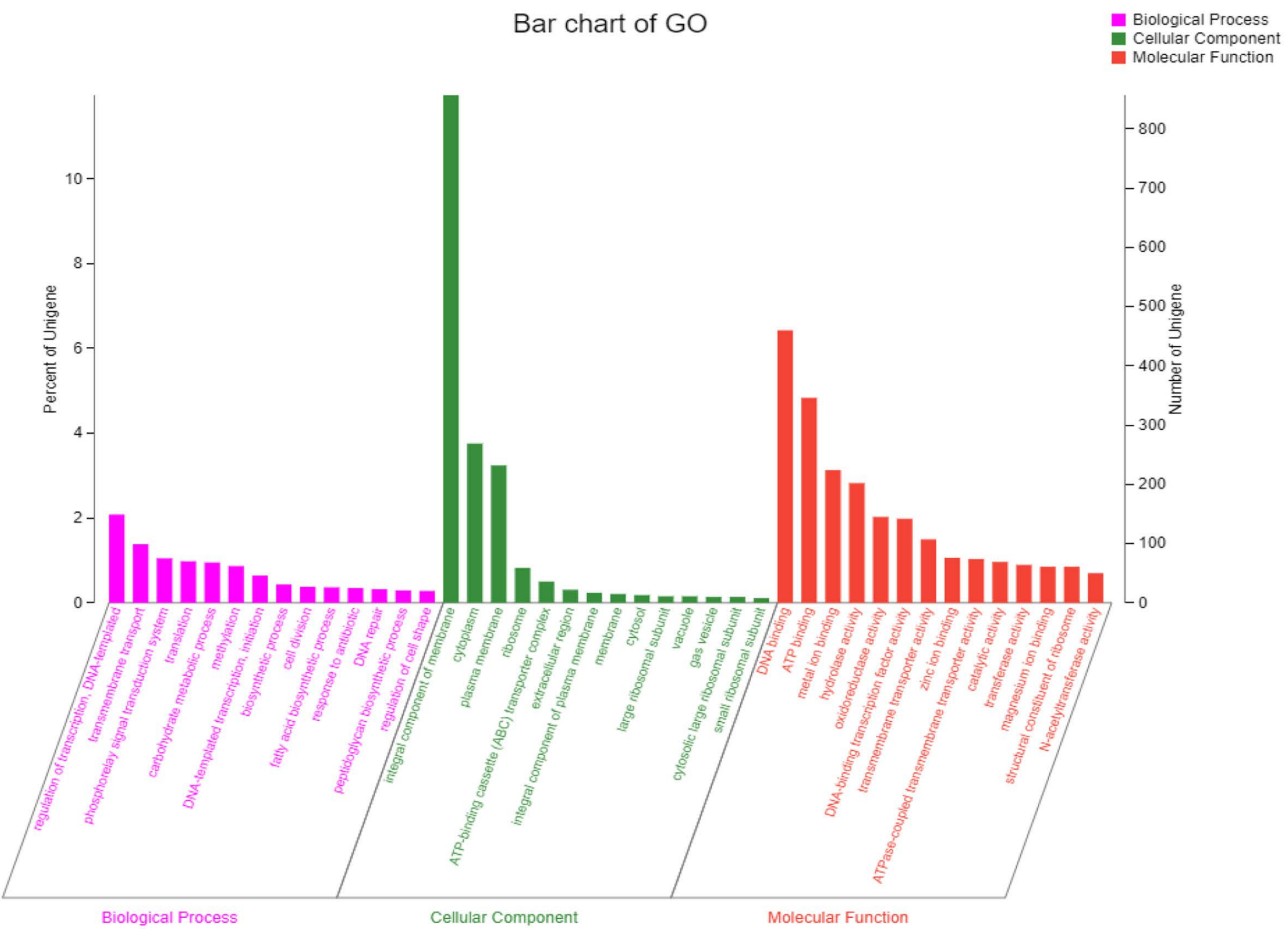

**Fig 5. GO functional classification map of *Streptomyces rochei* FE-3-1.**

**COG function classification: FE_3_1**

**Fig 6. COG functional classification map of *Streptomyces rochei* FE-3-1.**

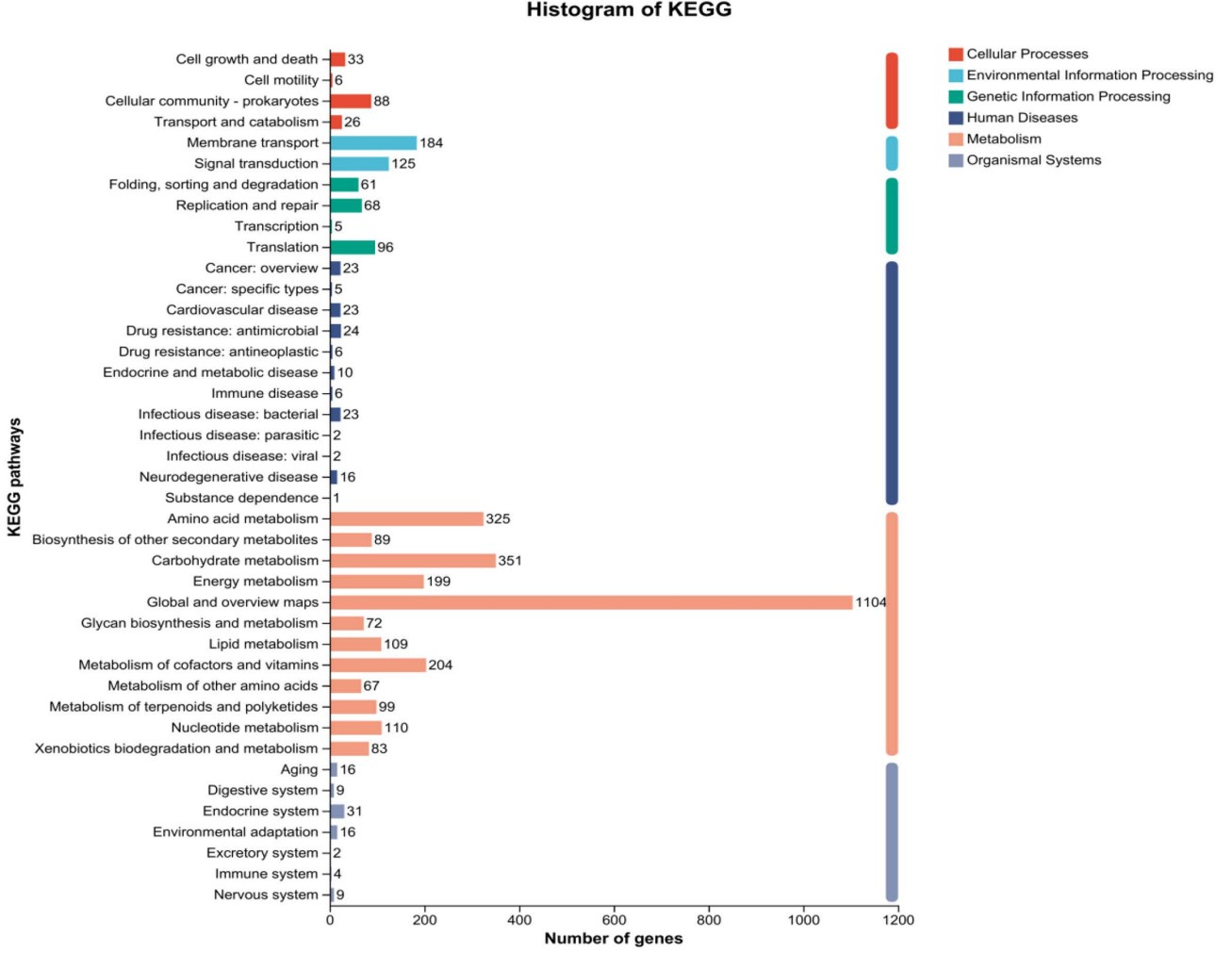

**Fig 7. KEGG classification of metabolic pathways map of *Streptomyces rochei* FE-3-1.**

### 3.7. Prediction of anti-*Pyricularia oryzae* activity

The strain FE-3–1 exhibited some antagonistic activity against *Pyricularia oryzae*, suggesting its potential as a biological control agent. Among the 31 gene clusters responsible for secondary metabolite production, the lipopeptides 8D1-1 and 8D1-2 from the NRPS/PKS gene cluster may play a major role in antagonizing *Pyricularia oryzae*. In the follow-up study, we will carry out the isolation, identification and function study of lipopeptides 8D1-1 and 8D1-2.

### 3.8. Molecular phylogenetic analysis of FE-3–1 and 8 other strains

To further clarify the taxonomic status of strain FE-3–1 at the species level, we compared its genome correlation index with that of 8 closely related model strains (S4 and S5 Tables in S1 File). Correlation analysis indicated a stronger correlation between different samples, with a highly significant difference (Fig 9A). The results also revealed that both the average nucleotide identity (ANI) and average amino acid identity (AAI) of strain FE-3–1 and *Streptomyces rochei* NRRL B-2410 were higher than the threshold values for species classification (ANI > 95%−96%, AAI > 95%−96%) (Fig 9B). Additionally, the genome G + C content of strain FE-3–1 and *Streptomyces rochei* NRRL

**Table 2. Putative proteins involved in metal resistance.**

| Metals | Gene name | Gene ID | Putative function | BLASTP analysis |
|---|---|---|---|---|
| Cadmium | copA | gene2642 | Cadmium-translocating P-type ATPase | WP_199577713.1 |
| | ctpJ | gene6062 | Cadmium-translocating P-type ATPase | WP_185912699.1 |
| Arsenic | arsB | gene0746 | Arsenic transporter | WP_136235124.1 |
| Zinc | znuA | gene2435 | Zinc ABC transporter substrate-binding protein | WP_127433667.1 |
| | ftsH | gene5105 | ATP-dependent zinc metalloprotease | WP_019328772.1 |
| Copper | ycnJ | gene3545 | Copper resistance protein | WP_127433257.1 |

**Table 3. Some genes related to the plant growth promotion within *Streptomyces rochei* FE-3-1 according to Prokka annotations.**

| Traits | Genes | Products |
|---|---|---|
| Nitrogen assimilation | nirB, nirD | Nitrite reductase |
| | narK, | Nitrate/nitrite transporter |
| | narI, narJ, narH, narG | Nitrate reductase |
| Phosphate solubilization | *rsb*U_P, *prp*C | Protein phosphatase |
| | phoD | Alkaline phosphatase D |
| | rhnA-cobC | RNase H/acid phosphatase |
| | maf | Nucleotide pyrophosphatase |
| | ppa | Inorganic diphosphatase |
| | gpmB | Histidine phosphatase |
| Potassium solubilization | kdpA, kdpB, kdpC | Potassium-transporting ATPase |
| | trkA | Potassium uptake protein |
| | cvrA | Potassium/proton antiporter |
| Iron sequestration | fhuD | Iron-siderophore ABC transporter substrate-binding protein |
| | – | Siderophore-interacting protein |
| Phytohormone synthesis | trpC | Indole-3-glycerol phosphate synthase |
| Biocatalyst | snpA | Zinc-dependent metalloprotease |
| | ftsH, htpX | Zinc metalloprotease |
| | hyaD | Hydrogenase maturation protease |
| | – | Lipase |
| | plc | Phospholipase |
| | pnbA | Carboxylesterase/lipase |
| | – | Chitinase |
| | amyA | Alpha-amylase |
| | katE | Catalase |
| | katG | Catalase/peroxidase |

B-2410 was found to be 72.51% and 72.50%, respectively, with a difference within the variation range of the same species (<1%). Furthermore, a syntenic analysis of FE-3–1 with *Streptomyces rochei* NRRL B-2410 genomes using Progressive Mauve Align showed a highly conserved synteny between them (Fig 10). Taking into consideration these genomic findings along with its morphological characteristics, we preliminarily identified strain FE-3–1 as belonging to *Streptomyces rochei*.

**Table 4. The biosynthetic gene clusters detected with AntiSMASH within the genome of *Streptomyces rochei* FE-3-1.**

| Cluster ID | Start | End | MIBiG accession | BGC Type | Most Similar Known Metabolies | % Identity | Gene No. |
|---|---|---|---|---|---|---|---|
| Cluster1 | 22348 | 106504 | BGC0001596 | T2PKS | fluostatins M-Q | 67 | 76 |
| Cluster2 | 175975 | 195904 | BGC0001294 | indole | 7-prenylisatin | 100 | 12 |
| Cluster3 | 199327 | 223208 | BGC0000664 | terpene | Isorenieratene | 100 | 21 |
| Cluster4 | 283217 | 491949 | BGC0000034 | NRPS | Candicidin | 95 | 67 |
| Cluster5 | 511227 | 531536 | BGC0000242 | terpene | lysolipin I | 4 | 21 |
| Cluster6 | 795716 | 816838 | BGC0001483 | indole | 5-isoprenylindole-3-carboxylate β-D-glycosyl ester | 33 | 23 |
| Cluster7 | 875531 | 898962 | BGC0000633 | terpene | carotenoid | 54 | 20 |
| Cluster8 | 1092239 | 1151522 | BGC0000432 | NRPS | streptothricin | 100 | 54 |
| Cluster9 | 1385222 | 1426329 | BGC0001065 | T3PKS | herboxidiene | 8 | 39 |
| Cluster10 | 2084186 | 2094585 | BGC0000853 | ectoine | ectoine | 100 | 10 |
| Cluster11 | 2965862 | 2976471 | BGC0000910 | melanin | melanin | 100 | 12 |
| Cluster12 | 3063027 | 3073950 | BGC0000940 | siderophore | desferrioxamin B/ desferrioxamine E | 83 | 9 |
| Cluster13 | 4094137 | 4137242 | BGC0000227 | NRPS-like | granaticin | 8 | 35 |
| Cluster14 | 4236271 | 4277228 | BGC0000914 | PKS-like | methylenomycin A | 9 | 44 |
| Cluster15 | 4375634 | 4395269 | BGC0000551 | lanthipeptide | SapB | 75 | 16 |
| Cluster16 | 4605927 | 4668548 | BGC0000806 | NRPS | phosphonoglycans | 5 | 61 |
| Cluster17 | 5327755 | 5348378 | BGC0000660 | terpene | albaflavenone | 100 | 20 |
| Cluster18 | 5386479 | 5455743 | BGC0000271 | T2PKS | spore pigment | 66 | 62 |
| Cluster19 | 5911783 | 5921934 | – | siderophore | – | – | 7 |
| Cluster20 | 6178528 | 6188937 | – | bacteriocin | – | – | 10 |
| Cluster21 | 6205131 | 6224912 | BGC0001181 | terpene | geosmin | 100 | 18 |
| Cluster22 | 6386962 | 6400136 | BGC0001732 | siderophore | paulomycin | 9 | 12 |
| Cluster23 | 6461777 | 6551885 | BGC0001370 | arylpolyene | lipopeptide 8D1-1/ lipopeptide 8D1-2 | 86 | 52 |
| Cluster24 | 6777423 | 6802165 | – | lanthipeptide | – | – | 28 |
| Cluster25 | 7154187 | 7180115 | BGC0000663 | terpene | hopene | 100 | 24 |
| Cluster26 | 7285725 | 7378488 | BGC0001785 | T1PKS | streptovaricin | 31 | 34 |
| Cluster27 | 7508419 | 7529451 | – | terpene | – | – | 20 |
| Cluster28 | 7540869 | 7551085 | BGC0000518 | bacteriocin | informatipeptin | 42 | 5 |
| Cluster29 | 7810082 | 7922797 | BGC0000031 | NRPS | borrelidin | 88 | 66 |
| Cluster30 | 8028118 | 8052713 | – | lanthipeptide | – | – | 20 |
| Cluster31 | 8193271 | 8247561 | BGC0000236 | lassopeptide | kinamycin | 31 | 57 |

## 3.9. Features of the core and pan-genomes

In order to better understand the genetic diversity of *Streptomyces*, we selected 8 of *Streptomyces* strains that are closely related to *Streptomyces rochei* FE-3–1 (Fig 9A), and studied the relationship between the size of the pan-genome and core genome and the number of genomes. With the addition of new strains, the number of core orthologous gene clusters consistently decreased, while the size of the pan-genome gradually increased (S1 and S2 Figs in S1 File). The gene content of in *Streptomyces rochei* FE-3–1 was compared with 8 other related reference strains. The comparison results showed that a total of 17,816 gene families were found in the 9 genomes, of which 2,920 genes constituted the core genome. The functional classes of the core gene families were further determined by homologous group (COG) attribution among all closely related species. The results showed that the core gene families were unevenly distributed across functional classes (Fig 6).

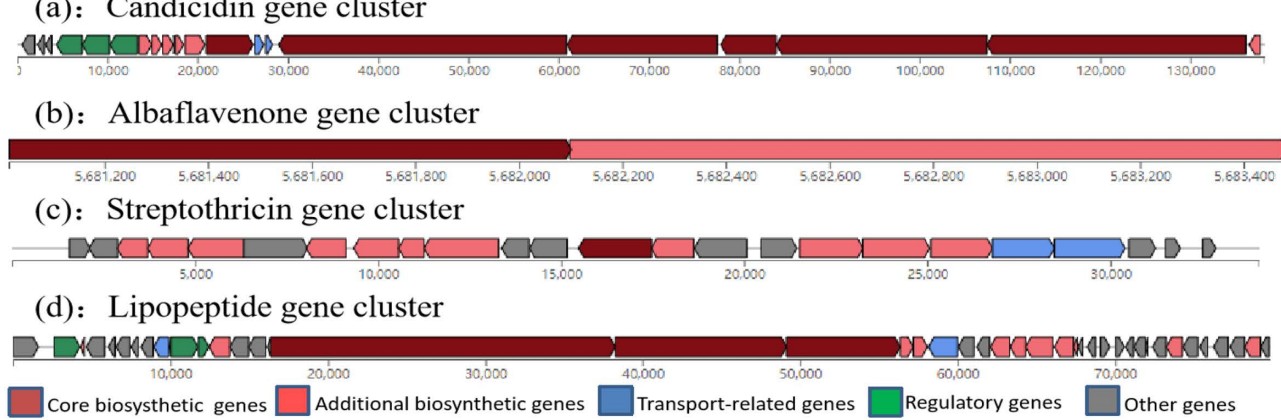

**Fig 8. Similarity to known antimicrobial metabolites biosynthetic gene clusters detected with antiSMASH in the genome of *Streptomyces rochei* FE-3-1: (A) Candicidin gene cluster; (B) Albaflavenone gene cluster; (C) Streptothricin gene cluster.** The characteristics of the clusters and gene types are indicated in the box.

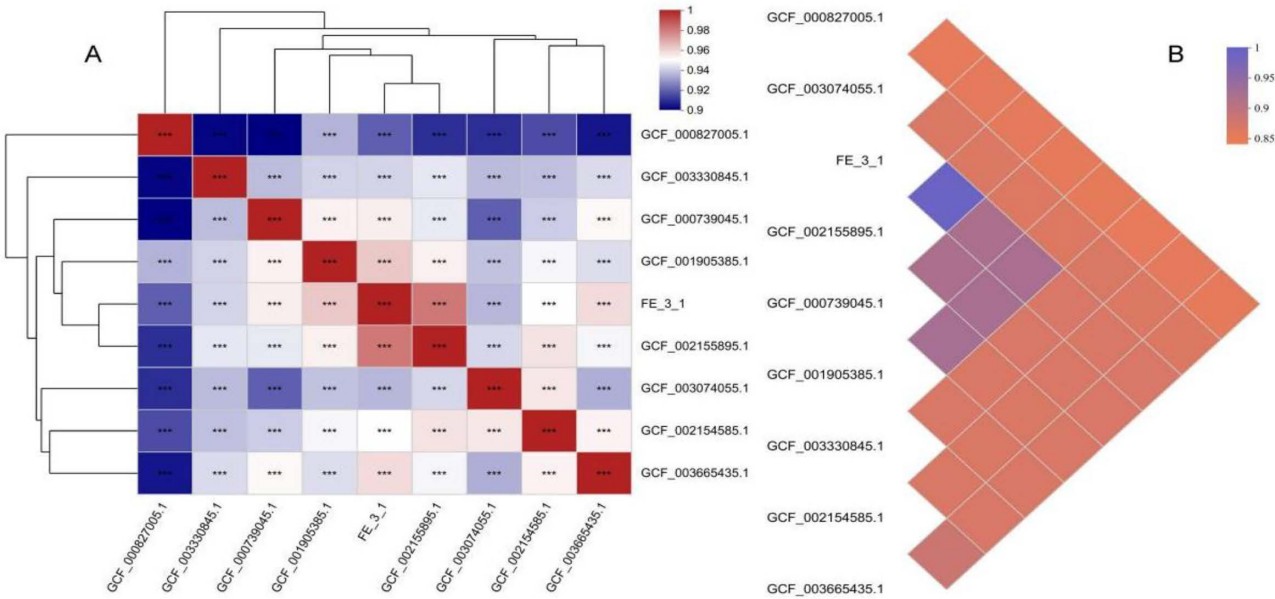

**Fig 9. Molecular phylogenetic analysis of FE-3-1 and 8 other strains of *Streptomyces rochei*.** (A) Correlation analysis of of 9 *Streptomyces* strains based on 16S rDNA sequence. 0.01 < p ≤ 0.05 *, 0.001 < p ≤ 0.01 **, p ≤ 0.001 ***. (B) ANI triangular heatmap of 9 *Streptomyces* strains. On the left is the sample name, and the squares of different colors indicate the average nucleotide similarity between the samples.

### 3.10. Comparative genomic analysis

In this study, the amino acid sequences of nine *Streptomyces* species involved were aligned using the OrthoMC, and a specific threshold (E-Value: 1e-5, Percent Identity Cutoff: 0, Markov Inflation Index: 1.5) was selected for similarity clustering in order to obtain homologous genes. A total of 8962 orthologous gene clusters were identified, with 2920 gene clusters being core orthologous gene clusters among the nine *Streptomyces* strains, and the number of homologous gene clusters co-existing in 2 or more samples was smaller than that in all samples (Fig 11A).Venn diagrams were

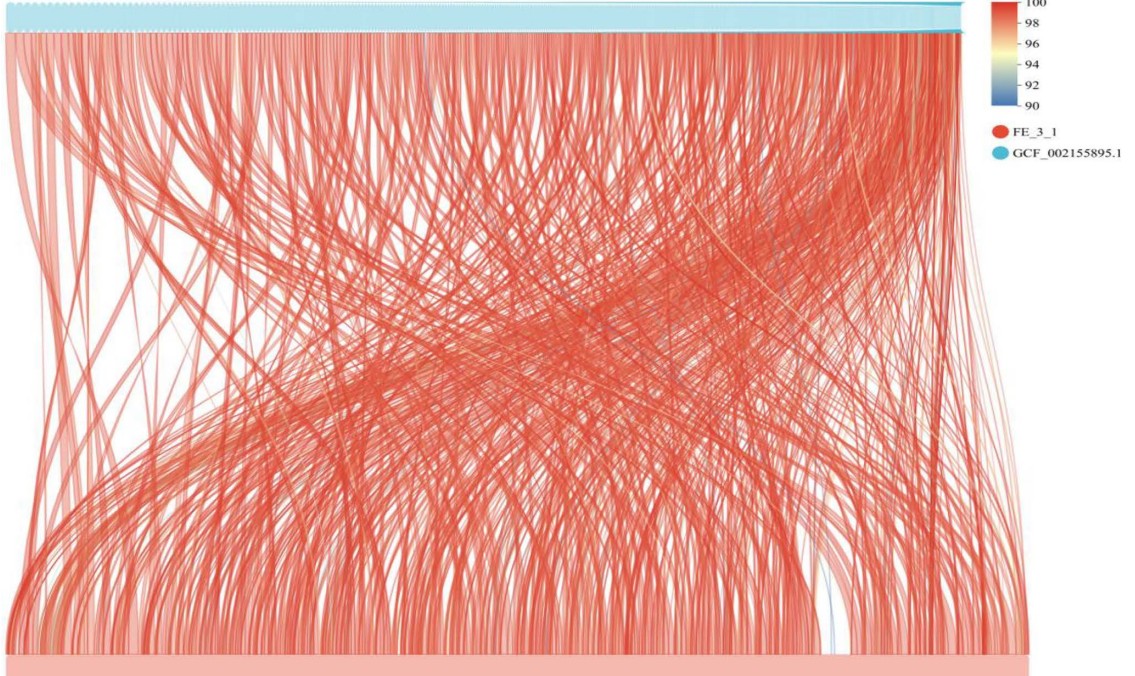

**Fig 10. Mauve synteny alignment analysis of the genome sequences of FE-3-1 and 8 other strains of *Streptomyces rochei*.** The upper and lower bars of color blocks represent two different genomes, and the regions of the two genomes are connected by lines, the color of which represents the degree of collinearity.

also used to intuitively display core and unique homologous genes between species. Among these, 244 clusters were found to be unique to *Streptomyces rochei* FE-3–1 and 543 clusters were unique to *Streptomyces rochei* NRRL B-2410 (GCF_002155895.1) (Fig 11B). The proportion of core, dispensable, and unique orthologous gene clusters varied across different samples' genomes (Fig 11C), and the number of new gene clusters gradually increased with the growing number of genomes (Fig 11D).

## 4. Discussion

With the rapid advancement of genome sequencing technology, large-scale genome sequencing continues to uncover the diverse natural products present in microbial resources. The biosynthetic potential of microorganisms has been significantly underestimated, ushering in a new era of microbial natural product mining post-genomics [23]. Genome mining utilizes bioinformatics analysis tools to predict secondary metabolic gene clusters for targeted product discovery, playing a crucial role in drug development [24]. Many microbial metabolites remain undiscovered through traditional low-throughput approaches because of the association of natural products with cryptic genes, which frequently demand external stimulation for microbes to generate. Genome mining expedites the identification of those cryptic and uncharacterized biosynthetic gene clusters accountable for synthesizing natural products, which are likely novel [24]. Furthermore, genomic information is invaluable for studying strain evolution, response mechanisms, and environmental adaptation [25].

Rhizobacteria play a vital role in promoting plant growth under various environmental conditions and also contribute to biocontrol by inhibiting plant pathogens through the production of secondary metabolites [26,27]. *Streptomyces* species are known for their rich source of bioactive substances [10]. In this study, *Streptomyces rochei* FE-3–1 isolated from rice

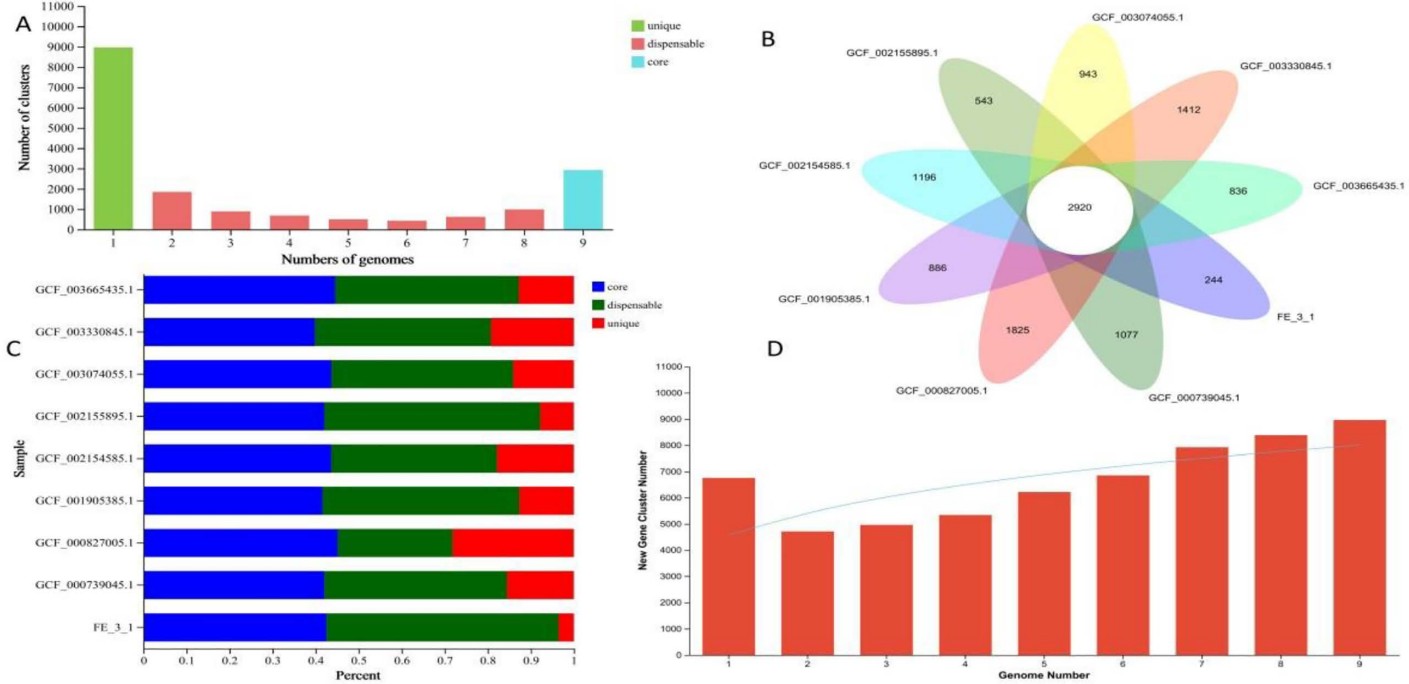

**Fig 11. Comparative genomic analysis of FE-3-1 and 8 other strains of *Streptomyces rochei*.** (A) Statistics of homologous genes in the genomes; (B) Flower plot displaying the numerous core orthologous gene clusters and specific orthologous gene clusters in 9 strains of *Streptomyces rochei.* Each petal represented a strain. The number of core orthologous gene clusters in shown in the center. The number in non-overlapping portions showed the numbers of strain-specific orthologous gene clusters. The strain name is located beside the petal; (C) Distribution of orthologous gene clusters in different samples; (D) Histogram and graph of the number of new gene clusters as a function of the number of genomes. A new gene is the number of gene clusters that are added when a genome is added for pan-genome analysis.

rhizosphere soil exhibited inhibitory effect on plant pathogen *Pyricularia oryzae* and holds potential for biocontrol applications as a rhizosphere *Streptomyces*.

Of all published related genomes of *Streptomyces rochei* FE-3–1, *Streptomyces rochei* NRRL B-2410 (GCF_002155895.1) shows the lowest genomic distance, and the strain is also isolated from soil and is the typical strain of *Streptomyces rochei*. The G+C content of *Streptomyces rochei* NRRL B-2410 was 72.5%, the number of genes was 7276, and the protein expression genes were 6727. Most of the members of the *Streptomyces rochei* group were reported as antibiotic-producing strains [28,29]. The genome of *Streptomyces rochei* NRRL B-2410 are understudied.

Comparative genomic analysis of nine distinct *Streptomyces* strains demonstrated that 244 unique gene clusters were specific to the strain FE-3–1. Among these unique homologous genes, such as nitrite reductase [EC:1.7.2.1], pyruvate carboxylase [EC:6.4.1.1], and polyphosphate kinase [EC:2.7.4.34], etc. The enzyme proteins encoded by these gene clusters play crucial roles in promoting growth, synthesizing secondary metabolites and tolerating external environmental stress. Moreover, these unique homologous genes may contribute to the strain's distinctive capabilities.

The KEGG pathways of *Streptomyces rochei* FE-3–1 show some genes that are beneficial to plants (Table 3). For example, reducing nitrate nitrogen in soil to ammonia is a beneficial ability for plants. It converts nitrate and nitrite compounds into ammonia compounds, which contributes to the availability of nitrogen for plants [30]. The strain can produce hydrogen sulfide through sulfur metabolism pathway (Table 5, Fig 12), which can combine with free heavy metal ions in soil to form residual sulfide, increase plant stress resistance and reduce the absorption and transport of heavy metals by plants [31]. Hydrogen sulfide can also act as a signaling molecule, promoting seed germination and seedling growth

**Table 5. Putative proteins involved in sulfate reduction.**

| Gene name | Gene ID | Putative function | BLASTP analysis |
|---|---|---|---|
| cysN | gene5529 | Sulfate adenylyltransferase subunit | WP_019326036.1 |
| cysD | gene5530 | Sulfate adenylyltransferase | WP_019326037.1 |
| cysC | gene5531 | Adenylyl-sulfate kinase | PVD05672.1 |
| cysH | gene5532 | Phosphoadenylyl-sulfate reductase | WP_125772742.1 |
| sirA | gene5534 | Sulfite reductase | WP_109204242.1 |

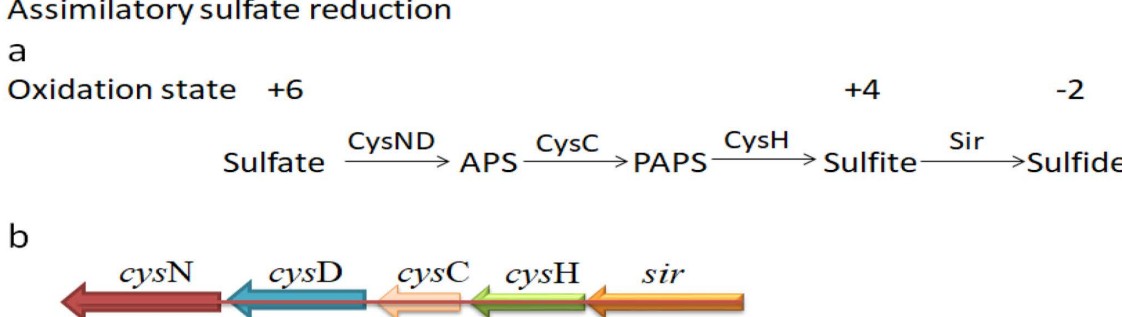

**Fig 12. Putative sulfur metabolism pathway and assimilatory sulfate reduction genes in *Streptomyces rochei* FE-3-1.** a Putative sulfur metabolism pathway. b Assimilatory sulfate reduction genes in *Streptomyces rochei* FE-3-1.

[32]. The sulfur metabolism pathway based on KEGG is illustrated in Fig 13. The strain has the ability to nitrogen assimilation, phosphate solubilization and potassium solubilization. By producing enzymes, the phosphorus that is difficult for plants to absorb and utilize is converted into absorbable and available phosphorus [33], and the insoluble potassium that plants cannot absorb is converted into soluble potassium [34]. By improving the utilization rate of nitrogen, phosphorus and potassium by plants, the strain promotes plant metabolism and regulates plant growth and reproduction. The strain FE-3–1 might function as a biocontrol agent through diverse mechanisms. The enhancement of plant vigor in the face of various biotic stresses, diseases, and pests is also regulated by an adequate supply of nitrogen, phosphate, and potassium.

Subsequent genomic analysis identified three siderophore gene clusters 12, 19, and 22 in the genome of *Streptomyces rochei* FE-3–1. Cluster 12 shows high similarity (83%) to the biosynthetic gene cluster of desferrioxamin B and desferrioxamine E in *Streptomyces coelicolor* A3(2) and contains 9 necessary genes responsible for the desferrioxamin B and desferrioxamine E biosynthesis. The siderophore produced by the strain can chelate iron ions in the environment, convert it into a form available to organisms, and promote plant growth. The siderophore produced by rhizosphere bacteria can also compete with plant pathogens for iron ions, thus inhibiting their growth and improving plant disease resistance. The siderophore can also chelate many harmful metal ions, reducing the damage of toxic metal ions to plants [35].

In order to destroy the components of cell wall of pathogenic fungal, the biosynthetic gene clusters of many functional microorganisms can produce hydrolytic enzymes, such as protease, lipase, amylase and chitinase, which is also an important mechanism for functional microorganisms to control plant pathogens [36]. Indeed, *Streptomyces rochei* FE-3–1 exhibited the potential to inhibit the growth of pathogenic fungus on agar plate, which was further confirmed and identified three protease, two lipase, two catalase, one chitinase and one amylase genes via genome mining in the genome of *Streptomyces rochei* FE-3–1 (Table 3).

**Fig 13. The sulfur metabolism is illustrated using a KEGG pathway diagram.** The proteins highlighted in red boxes were identified in strain FE-3-1. Numbers indicate E.C. numbers of enzymes.

The soil heavy metal pollution constitutes a crucial limiting factor for the survival of the strain in the environment. Bacteria have evolved several mechanisms to resist and remediate heavy metals for their own survival [31]. Certain bacterial metabolic mechanisms specifically exist to immobilize and reduce the bioavailability of heavy metals. Immobilization can be accomplished through sorption to cell components and exopolymers, transport, intracellular sequestration, or precipitation of metal ions as insoluble organic and inorganic compounds [37]. Based on the outcomes of genome annotation, the strain FE-3–1 possesses multiple putative functional proteins related to various heavy metals. These resistances to heavy metals enhance the strain FE-3–1's capability to survive in extreme environments and lay a foundation for its colonization of the rhizosphere and promotion of plant growth.

Table 6. Commercialized *streptomyces* based biopesticides [19].

| Biocontrol agent(s) | Active ingredient | Target pathogen(s) | Biocontrol mechanism | Country |
|---|---|---|---|---|
| Actinovate | streptomyces lydicus WYEC108 | Fusarium spp., Rhizoctonia spp., Pythium spp., Phytophthora spp., Erisiphe spp., Sphaeroteca spp., Laveillula spp., Sclerotinia spp. | Antibiosis and hyperparasitism | European Union |
| Rhizovit | Streptomyces rimosus | Pythium spp., Fusarium spp., Phomopsis spp., Phytophthora spp., R. solani, A. brassicola, Botrytis spp., Fusarium spp. | Antibiosis | – |
| Mycostop | Streptomyces griseoviridis K61 | Ceratocystis radicicola, Alternaria spp., Rhizoctonia solani, Fusarium spp., Phytophthora spp., Pythium spp. | Competition, hyperparasitism, and antibiosis | Canada |

The antiSMASH results showed that there were 31 biosynthetic gene clusters. The number of NRPS and terpene-type biosynthetic gene clusters was the largest. PKS is significantly present in both terrestrial and aquatic environments and is closely related to the biosynthesis of antimicrobial metabolites in *actinomycetes*, especially *Streptomyces* [38]. Five of PKS and PKS-like gene clusters were found in the genome of *Streptomyces rochei* FE-3–1. At least three of the gene clusters detected in the genome of *Streptomyces rochei* FE-3–1 are associated with currently known potent antimicrobials. Strepto-thricin is an N-glucoside antibiotic, which exists in different *Streptomyces* and is one of the earliest antibiotics isolated from *Streptomyces* and has broad-spectrum activity against both Gram-positive, Gram-negative bacteria and pathogenic fungi. It inhibits protein biosynthesis in prokaryotic cells by suppressing polypeptide synthesis through ribosome interactions [39]. The candicidin complex was initially isolated from *Streptomyces griseus* IMRU3570. It demonstrates a potent inhibitory activity against fungi. The candicidin complex was regarded as the prototype of aromatic polyene macrolides. Similar to other polyketide polyenes, candicidin can interact with ergosterol existing in the membranes of fungi to form a transmembrane channel, resulting in $K^+$ leakage and causing cell death [40]. Albaflavenone was initially isolated from *Streptomyces albidoflavus* and is recognized for exhibiting antibacterial properties. The biosynthetic pathway of albaflavenone has recently been elucidated in *Streptomyces coelicolor* A3(2) [41]. Lipopeptide antibiotics represent the most recent addition to the antibiotic family, with molecular weights typically ranging from 1000−1600 Da. They are synthesized via a non-ribosomal multi-enzyme biosynthesis pathway [42]. Surfactin, iturin, and fengycin are exemplary lipid peptides known for their intricate effects on plant pathogenic fungi [43]. Existing literature also contains relevant reports on lipopeptides antagonizing Pyricularia oryzae [44–46]. These results may explain the extensive and effective bacteriostatic activity of *Streptomyces rochei* FE-3–1. Other biosynthetic gene clusters are associated with other active metabolites, including isoprene, which has antioxidant effects. Ectoine, which is resistant to salt stress. Siderophore, which has strong chelating ability to iron. melanin, which can resist environmental stress, and borrelidin, which has herbicidal activity.

Numerous commercial biocontrol agents derived from diverse *Streptomyces* strains, including Actinovate, Rhizovit, and Mycostop, exhibit resistance to a wide range of pathogens (Table 6) [19]. While this study demonstrates the potent *in vitro* antagonism of *Streptomyces rochei* FE-3–1 metabolites against *Pyricularia oryzae*, a direct quantitative comparison of inhibition rates to commercial products like Actinovate (*Streptomyces lydicus*), Rhizovit (*Streptomyces* sp.), or Mycostop (*Streptomyces griseoviridis*) requires further bioassay against a common pathogen panel. However, the genomic analysis reveals FE-3–1 possesses significant potential comparable to these commercially exploited strains. Its genome harbors 31 biosynthetic gene clusters, at least four linked to known potent antimicrobials. This genomic richness suggests FE-3–1 has the inherent capacity to produce a diverse arsenal of bioactive compounds, potentially explaining its observed activity

and supporting its potential to resist a wide range of plant pathogens. This preliminary genomic evidence positions FE-3–1 as a strong candidate for development into novel biofertilizers or biopesticides, warranting further exploration of its metabolite production.

## 5. Conclusions

This study isolated a *Streptomyces* strain, FE-3–1, from the rice rhizosphere, demonstrating significant inhibitory activity against the plant pathogen *Pyricularia oryzae*. Morphological and phylogenetic analysis identified the strain as *Streptomyces rochei*. Whole genome sequencing revealed a substantial biosynthetic potential, anchored by 31 biosynthetic gene clusters including at least four associated with known antimicrobials. This high-quality genome sequence provides a crucial resource for comparative actinobacterial genomics. Genome mining confirmed the strain's strong capacity for secondary metabolite production, highlighting its value as a source of bioactive compounds. Collectively, these findings provide a robust theoretical foundation for developing *Streptomyces rochei* FE-3–1 into effective biofertilizers or biopesticides. Furthermore, the genomic insights facilitate the targeted mining of related, underexplored actinobacteria, paving the way for discovering next-generation biocontrol strains with novel or enhanced modes of action against diverse plant pathogens.

## Supporting information

**S1 File.  Supplementary materials.**
(DOC)

## Author contributions

**Conceptualization:** Dongxia Du.

**Data curation:** Shiping Shan.

**Funding acquisition:** Shiping Shan.

**Methodology:** Dongxia Du.

**Writing – original draft:** Dongxia Du.

**Writing – review & editing:** Dongxia Du, Zhuo Yi, Shiping Shan, Shuaishuai Gao, Mengyuan Yu, Bin Wang.

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
