## [Decision Letter · Decision Letter 0]

8 Oct 2024

PONE-D-24-32365

Isolation, identification, and whole-genome sequencing of Streptomyces rochei FE-3-1 against Pyricularia oryzae

PLOS ONE

Dear Dr. Shan,

Thank you for submitting your manuscript to PLOS ONE. After careful consideration, we feel that it has merit but does not fully meet PLOS ONE’s publication criteria as it currently stands. Therefore, we invite you to submit a revised version of the manuscript that addresses the points raised during the review process.

We look forward to receiving your revised manuscript.

Kind regards,

Yunzhou Li

Academic Editor

PLOS ONE

Journal Requirements:

Reviewers' comments:

Reviewer's Responses to Questions

**Comments to the Author**

1. Is the manuscript technically sound, and do the data support the conclusions?

Reviewer #1: Yes

2. Has the statistical analysis been performed appropriately and rigorously? 

Reviewer #1: No

3. Have the authors made all data underlying the findings in their manuscript fully available?

Reviewer #1: Yes

4. Is the manuscript presented in an intelligible fashion and written in standard English?

Reviewer #1: Yes

5. Review Comments to the Author

Reviewer #1: After carefully reviewing the manuscript titled "Isolation, Identification, and Whole-Genome Sequencing of Streptomyces rochei FE-3-1 Against Pyricularia oryzae" (Manuscript ID: PONE-D-24-32365), I find the research both relevant and timely, particularly in terms of its implications for biocontrol and antibiotic discovery. The genome analysis and biocontrol potential of Streptomyces rochei FE-3-1 represent significant contributions to the field.

However, before recommending the manuscript for publication, I would like to suggest a few points for revision and clarification. Below, I outline the questions and recommendations that could help improve the depth and clarity of the manuscript:

1. Lines 87-99: You mention the potential use of Streptomyces rochei FE-3-1 as a biopesticide or biofertilizer. Could you expand on the next steps in product development or field trials? This would help position the findings for practical, real-world applications.

2. Lines 111-121: Please provide more details on the statistical methods used to validate the inhibition rates observed in the antagonistic activity assay against Pyricularia oryzae. Including specific statistical tests will add rigor to the data and help readers assess the reliability of the results.

Additionally, have you assessed the ability of Streptomyces rochei FE-3-1 to control Pyricularia oryzae-caused disease in actual rice plants, beyond the in vitro antagonistic assays? If so, please provide detailed information on these trials, as it would greatly enhance the practical relevance of your findings.

3. Lines 229-237: The identification of heavy metal resistance genes is intriguing, but it would be beneficial to discuss how these genes enhance the strain’s survival and effectiveness in agricultural environments that are subject to pollution or heavy metal contamination.

4. Lines 240-253: Could you clarify how the genes related to plant growth promotion, such as those for phosphorus and potassium solubilization, enhance the biocontrol efficiency of Streptomyces rochei FE-3-1? Understanding the synergy between plant growth and pathogen inhibition would be valuable.

5. Lines 257-274: The manuscript identifies several biosynthetic gene clusters, such as those for candicidin and streptothricin, which are related to antimicrobial activity. Expanding on how these clusters contribute to the biocontrol properties of Streptomyces rochei FE-3-1 would strengthen the link between the genomic data and the biological functions of the strain.

6. Lines 257-277: It would be helpful to include a comparison of the identified secondary metabolites with known commercial biocontrol agents in terms of their effectiveness and range of activity. This could give a clearer perspective on the potential competitive advantage of Streptomyces rochei FE-3-1.

7. Lines 333-362: The manuscript mentions unique homologous genes found in Streptomyces rochei FE-3-1. It would be beneficial to discuss how these genes might contribute to the strain’s unique abilities or adaptations, especially in comparison with other Streptomyces species.

8. Lines 364-374: Finally, your study contributes significantly to microbial genome mining. Could you emphasize how your findings advance the field of genome-guided discovery of biocontrol agents and antibiotics?

9. Introducing biocontrol agents into agricultural systems carries some risk. Have you considered any potential environmental impacts or risks associated with using Streptomyces rochei FE-3-1? If so, how might these be mitigated?

6. PLOS authors have the option to publish the peer review history of their article (what does this mean? ). If published, this will include your full peer review and any attached files.

**Do you want your identity to be public for this peer review?** For information about this choice, including consent withdrawal, please see our Privacy Policy .

Reviewer #1: No

---

## [Author Response · Author response to Decision Letter 1]

16 Oct 2024

Dear editors,

Thank you very much for your E-mail with regard to our manuscript (PONE-D-24-32365) together with the comments from the reviewers. The comments from the reviewers were very helpful. We have responded each comment from the reviewers in a separate letter and taken all of these comments into account in preparing the revised manuscript. All of updates in the manuscript were marked via changing the font colour of the text. We believe that manuscript has been revised satisfactorily and hope it will be accepted for publication in PLOS ONE.

We thank again the reviewers for the helpful comments. Should you require any further information, please do not hesitate to ask.

Thanks very much for your attention and consideration.

Sincerely yours

Shiping Shan

---

## [Decision Letter · Decision Letter 1]

18 Dec 2024

PONE-D-24-32365R1Isolation, identification, and whole-genome sequencing of Streptomyces rochei FE-3-1 against Pyricularia oryzaePLOS ONE

Dear Dr. Shan,

Thank you for submitting your manuscript to PLOS ONE. After careful consideration, we feel that it has merit but does not fully meet PLOS ONE’s publication criteria as it currently stands. Therefore, we invite you to submit a revised version of the manuscript that addresses the points raised during the review process.

We look forward to receiving your revised manuscript.

Kind regards,

Yunzhou Li

Academic Editor

PLOS ONE

Journal Requirements:

Reviewers' comments:

Reviewer's Responses to Questions

**Comments to the Author**

1. If the authors have adequately addressed your comments raised in a previous round of review and you feel that this manuscript is now acceptable for publication, you may indicate that here to bypass the “Comments to the Author” section, enter your conflict of interest statement in the “Confidential to Editor” section, and submit your "Accept" recommendation.

Reviewer #1: All comments have been addressed

2. Is the manuscript technically sound, and do the data support the conclusions?

Reviewer #1: Yes

3. Has the statistical analysis been performed appropriately and rigorously? 

Reviewer #1: Yes

4. Have the authors made all data underlying the findings in their manuscript fully available?

Reviewer #1: Yes

5. Is the manuscript presented in an intelligible fashion and written in standard English?

Reviewer #1: Yes

6. Review Comments to the Author

Reviewer #1: The author has made corrections according to the suggestions, but minor errors remain, such as the need to italicize the scientific name Streptomyces. It is therefore recommended that the author thoroughly check the manuscript for accuracy.

7. PLOS authors have the option to publish the peer review history of their article (what does this mean? ). If published, this will include your full peer review and any attached files.

**Do you want your identity to be public for this peer review?** For information about this choice, including consent withdrawal, please see our Privacy Policy .

Reviewer #1: No

---

## [Author Response · Author response to Decision Letter 2]

18 Dec 2024

Q-1: The author has made corrections according to the suggestions, but minor errors remain, such as the need to italicize the scientific name Streptomyces. It is therefore recommended that the author thoroughly check the manuscript for accuracy.

R-1: All the authors thoroughly reviewed the entire manuscript and revised all scientific names to be presented in italics on Page23 Line403, Page29 Line518, Page31 Line561, Page36 Line661-662, Page37 Line698. and the changes have been highlighted in red.

---

## [Decision Letter · Decision Letter 2]

2 Jun 2025

PONE-D-24-32365R2Isolation, identification, and whole-genome sequencing of Streptomyces rochei FE-3-1 against Pyricularia oryzaePLOS ONE

Dear Dr. Shan,

Thank you for submitting your manuscript to PLOS ONE. After careful consideration, we feel that it has merit but does not fully meet PLOS ONE’s publication criteria as it currently stands. Therefore, we invite you to submit a revised version of the manuscript that addresses the points raised during the review process.

After carefully reviewing the manuscript titled "Isolation, Identification, and Whole-Genome Sequencing of Streptomyces rochei FE-3-1 Against Pyricularia oryzae" (Manuscript ID: PONE-D-24-32365), I find the research both relevant and timely, particularly in terms of its implications for biocontrol and antibiotic discovery. The genome analysis and biocontrol potential of Streptomyces rochei FE-3-1 represent significant contributions to the field.

However, before recommending the manuscript for publication, I would like to suggest a few points for revision and clarification. Below, I outline the questions and recommendations that could help improve the depth and clarity of the manuscript:

1. Lines 87-99: You mention the potential use of Streptomyces rochei FE-3-1 as a biopesticide or biofertilizer. Could you expand on the next steps in product development or field trials? This would help position the findings for practical, real-world applications.

2. Lines 111-121: Please provide more details on the statistical methods used to validate the inhibition rates observed in the antagonistic activity assay against Pyricularia oryzae. Including specific statistical tests will add rigor to the data and help readers assess the reliability of the results.

Additionally, have you assessed the ability of Streptomyces rochei FE-3-1 to control Pyricularia oryzae-caused disease in actual rice plants, beyond the in vitro antagonistic assays? If so, please provide detailed information on these trials, as it would greatly enhance the practical relevance of your findings.

3. Lines 229-237: The identification of heavy metal resistance genes is intriguing, but it would be beneficial to discuss how these genes enhance the strain’s survival and effectiveness in agricultural environments that are subject to pollution or heavy metal contamination.

4. Lines 240-253: Could you clarify how the genes related to plant growth promotion, such as those for phosphorus and potassium solubilization, enhance the biocontrol efficiency of Streptomyces rochei FE-3-1? Understanding the synergy between plant growth and pathogen inhibition would be valuable.

5. Lines 257-274: The manuscript identifies several biosynthetic gene clusters, such as those for candicidin and streptothricin, which are related to antimicrobial activity. Expanding on how these clusters contribute to the biocontrol properties of Streptomyces rochei FE-3-1 would strengthen the link between the genomic data and the biological functions of the strain.

6. Lines 257-277: It would be helpful to include a comparison of the identified secondary metabolites with known commercial biocontrol agents in terms of their effectiveness and range of activity. This could give a clearer perspective on the potential competitive advantage of Streptomyces rochei FE-3-1.

7. Lines 333-362: The manuscript mentions unique homologous genes found in Streptomyces rochei FE-3-1. It would be beneficial to discuss how these genes might contribute to the strain’s unique abilities or adaptations, especially in comparison with other Streptomyces species.

8. Lines 364-374: Finally, your study contributes significantly to microbial genome mining. Could you emphasize how your findings advance the field of genome-guided discovery of biocontrol agents and antibiotics?

9. Introducing biocontrol agents into agricultural systems carries some risk. Have you considered any potential environmental impacts or risks associated with using Streptomyces rochei FE-3-1? If so, how might these be mitigated?==============================

We look forward to receiving your revised manuscript.

Kind regards,

Sujata Singh Yadav

Academic Editor

PLOS ONE

**Journal Requirements:**

Reviewers' comments:

Reviewer's Responses to Questions

**Comments to the Author**

1. If the authors have adequately addressed your comments raised in a previous round of review and you feel that this manuscript is now acceptable for publication, you may indicate that here to bypass the “Comments to the Author” section, enter your conflict of interest statement in the “Confidential to Editor” section, and submit your "Accept" recommendation.

Reviewer #1: All comments have been addressed

2. Is the manuscript technically sound, and do the data support the conclusions?

Reviewer #1: Yes

3. Has the statistical analysis been performed appropriately and rigorously? 

Reviewer #1: Yes

4. Have the authors made all data underlying the findings in their manuscript fully available?

Reviewer #1: Yes

5. Is the manuscript presented in an intelligible fashion and written in standard English?

Reviewer #1: Yes

6. Review Comments to the Author

**Reviewer #1: ** The author has revised the MS “Isolation, identification, and whole-genome sequencing of Streptomyces rochei FE-3-1 against Pyricularia oryzae” in several aspects as suggested.

7. PLOS authors have the option to publish the peer review history of their article (what does this mean? ). If published, this will include your full peer review and any attached files.

**Do you want your identity to be public for this peer review?** For information about this choice, including consent withdrawal, please see our Privacy Policy .

Reviewer #1: No

---

## [Author Response · Author response to Decision Letter 3]

6 Jun 2025

Dear editors,

Thank you very much for your E-mail with regard to our manuscript (PONE-D-24-32365R2) together with the comments from the reviewers. The comments from the reviewers were very helpful. We have responded each comment from the reviewers in a separate letter and taken all of these comments into account in preparing the revised manuscript. All of updates in the manuscript were marked via changing the font colour of the text. We believe that manuscript has been revised satisfactorily and hope it will be accepted for publication in PLOS ONE.

We thank again the reviewers for the helpful comments. Should you require any further information, please do not hesitate to ask.

Thanks very much for your attention and consideration.

Sincerely yours

Shiping Shan

---

## [Decision Letter · Decision Letter 3]

13 Aug 2025

PONE-D-24-32365R3Isolation, identification, and whole-genome sequencing of Streptomyces rochei FE-3-1 against Pyricularia oryzaePLOS ONE

Dear Dr. Shan,

Thank you for submitting your manuscript to PLOS ONE. After careful consideration, we feel that it has merit but does not fully meet PLOS ONE’s publication criteria as it currently stands. Therefore, we invite you to submit a revised version of the manuscript that addresses the points raised during the review process.

We look forward to receiving your revised manuscript.

Kind regards,

Chetan Keswani, Ph.D.

Academic Editor

PLOS ONE

Journal Requirements:

Reviewers' comments:

Reviewer's Responses to Questions

**Comments to the Author**

1. If the authors have adequately addressed your comments raised in a previous round of review and you feel that this manuscript is now acceptable for publication, you may indicate that here to bypass the “Comments to the Author” section, enter your conflict of interest statement in the “Confidential to Editor” section, and submit your "Accept" recommendation.

Reviewer #1: All comments have been addressed

Reviewer #2: (No Response)

2. Is the manuscript technically sound, and do the data support the conclusions?

Reviewer #1: Yes

Reviewer #2: Yes

3. Has the statistical analysis been performed appropriately and rigorously? 

Reviewer #1: Yes

Reviewer #2: Yes

4. Have the authors made all data underlying the findings in their manuscript fully available?

Reviewer #1: Yes

Reviewer #2: Yes

5. Is the manuscript presented in an intelligible fashion and written in standard English?

Reviewer #1: Yes

Reviewer #2: Yes

6. Review Comments to the Author

Reviewer #1: The author has addressed the suggested revisions in multiple sections of the manuscript titled 'Isolation, identification, and whole-genome sequencing of Streptomyces rochei FE-3-1 against Pyricularia oryzae.

Reviewer #2: This manuscript reports the isolation of Streptomyces rochei FE-3-1 from rice rhizosphere, its antifungal activity against rice blast fungus and a detailed genomic analysis elucidating its biocontrol and plant growth-promoting potential. The study includes classical microbiological approaches, genome mining, and comparative genomics including comparisons with related Streptomyces species. The use of whole-genome sequencing, functional annotation, and the practical insight into biocontrol product development align well its aim.

Relevance and novelty of this paper.

The biocontrol of rice blast is a problem of global agricultural importance. The study addresses a significant challenge in sustainable crop protection by focusing on a native rhizosphere Streptomyces strain. The complete genome sequence for S. rochei FE-3-1 is newly reported, including genome mining for secondary metabolites and resistance genes, and there is thoughtful discussion of the translational path toward product development (biopesticide/biofertilizer).

Both the main text and the cover letter clarify that the whole genome sequence and detailed comparative genomics of S. rochei FE-3-1 are first reported in this study. The NCBI accession numbers are provided for verification. There is no indication of prior publication of these data elsewhere.

Experimental design and logic.

The experimental provides a comprehensive workflow. The work combines phenotype assays, genome sequencing, mining for beneficial traits, and comparative genomics. All relevant data (accession numbers for GenBank, etc.) are made available in accordance with PLOS ONE standards.

Experimental replication and statistical methods are clearly described for laboratory inhibition data, and the connection between detected biosynthetic gene clusters and mechanisms of pathogen inhibition is explicitly made.

Standard microbiological, genetic, and bioinformatics approaches (antiSMASH, Prokka, phylogenetic analysis) are applied. The main claims (FE-3-1's antagonism in vitro, genomic potential for biocontrol and growth-promotion, biosynthetic gene clusters) are directly supported by experimental/genomic data.

Though I believe that limitations of this study, e.g., using in vitro data only, should be acknowledged more clearly. The environmental risk assessment is limited which is another weak point.

1. The supplementary table (S6) provides an overview of commercial benchmarks. It might be helpful to briefly discuss in the main text how the observed inhibition rates and genomic potential of FE-3-1 compare quantitatively, where possible, to those products. Also it is unclear which of references 1-4 relates to respective preparation.

2. English: there are minor issues with grammar and phrasing throughout the text. A careful editing may be necessary. Although my judgment may be subjective due to the fact that I’m not a native speaker.

Suggestions

• If possible, reference supporting data (statistical outputs, graphical results) directly in the main text rather than relying on supplementary figures.

• In the discussion, clearly differentiate between what is demonstrated (in vitro/in silico) and what is still hypothetical or future work (in planta, field results, compound isolation, commercial development).

• Reinforce the conclusion with perspective on how this genome resource could contribute to the mining of new actinobacterial biocontrol strains.

Overall conclusion:

Minor revisions are recommended

7. PLOS authors have the option to publish the peer review history of their article (what does this mean? ). If published, this will include your full peer review and any attached files.

**Do you want your identity to be public for this peer review?** For information about this choice, including consent withdrawal, please see our Privacy Policy .

Reviewer #1: No

Reviewer #2: No

---

## [Author Response · Author response to Decision Letter 4]

13 Aug 2025

To the Comments of Reviewer

Q-1: The supplementary table (S6) provides an overview of commercial benchmarks. It might be helpful to briefly discuss in the main text how the observed inhibition rates and genomic potential of FE-3-1 compare quantitatively, where possible, to those products. Also it is unclear which of references 1-4 relates to respective preparation.

R-1: The following contents have been added on Page 30-31 Line 539-554, and the changes have been highlighted in red.

While this study demonstrates the potent in vitro antagonism of Streptomyces rochei FE-3-1 metabolites against Pyricularia oryzae, a direct quantitative comparison of inhibition rates to commercial products like Actinovate (Streptomyces lydicus), Rhizovit (Streptomyces sp.), or Mycostop (Streptomyces griseoviridis) requires further bioassay against a common pathogen panel. However, the genomic analysis reveals FE-3-1 possesses significant potential comparable to these commercially exploited strains. Its genome harbors 31 biosynthetic gene clusters, at least four linked to known potent antimicrobials. This genomic richness suggests FE-3-1 has the inherent capacity to produce a diverse arsenal of bioactive compounds, potentially explaining its observed activity and supporting its potential to resist a wide range of plant pathogens. This preliminary genomic evidence positions FE-3-1 as a strong candidate for development into novel biofertilizers or biopesticides, warranting further exploration of its metabolite production.

References 1 to 4 have been respectively cited at appropriate positions in the main text.

The reference 1 has been cited on Page 3 Line 45, and the changes have been highlighted in red.

The reference 2 has been cited on Page 3 Line 48, and the changes have been highlighted in red.

The reference 3 has been cited on Page 3 Line 51, and the changes have been highlighted in red.

The reference 4 has been cited on Page 3 Line 53, and the changes have been highlighted in red.

The reference 5 has been cited on Page 3 Line 56, and the changes have been highlighted in red.

Q-2: English: there are minor issues with grammar and phrasing throughout the text. A careful editing may be necessary. Although my judgment may be subjective due to the fact that I’m not a native speaker.

R-2: All of us have revised the whole manuscript and corrected the grammatical and phrasing errors.

The word "has" has been changed to "have" on Page 3 Line 47, and the changes have been highlighted in red.

The word "bacteria" has been changed to "bacterial species" on Page 3 Line 58, and the changes have been highlighted in red.

The sentence "They also play a crucial role as growth promotion and biocontrol agents, and their use in agriculture is becoming increasingly widespread" has been changed to "They also play a crucial role as agents for growth promotion and biocontrol, with their application in agriculture becoming increasingly prevalent" on Page 3 Line 58-60, and the changes have been highlighted in red.

The phrase "Prokka annotated" has been changed to "Prokka- annotated" on Page 8 Line 149-150, and the changes have been highlighted in red.

The word "identify" has been changed to " identified" on Page 19 Line 309, and the changes have been highlighted in red.

The word "with" has been changed to " and" on Page 20 Line 328, and the changes have been highlighted in red.

The phrase "S6 Table" has been changed to "Table 6" on Page 29 Line 520, and the changes have been highlighted in red.

Q-3: If possible, reference supporting data (statistical outputs, graphical results) directly in the main text rather than relying on supplementary figures..

R-3: The supporting reference data S6 has been relocated to the main text and is now designated as Table 6.

Q-4: In the discussion, clearly differentiate between what is demonstrated (in vitro/in silico) and what is still hypothetical or future work (in planta, field results, compound isolation, commercial development).

R-4: Our study demonstrates in vitro that Streptomyces rochei FE-3-1 inhibits the plant pathogen Pyricularia oryzae on agar plates, confirming its biocontrol potential. In silico genomic analysis revealed 31 biosynthetic gene clusters (BGCs), including 5 PKS/PKS-like clusters, and identified genes encoding hydrolytic enzymes (proteases, lipases, amylase, chitinase) potentially involved in fungal cell wall degradation. Comparative genomics showed 244 unique gene clusters in FE-3-1, including those encoding nitrite reductase, pyruvate carboxylase, and polyphosphate kinase, which may contribute to environmental stress tolerance and secondary metabolism. KEGG pathway analysis in silico further identified genes linked to plant-beneficial functions: nitrogen assimilation (nitrate reduction to ammonia), phosphate/potassium solubilization, and sulfur metabolism (hydrogen sulfide production via enzymes CysN, CysD, CysC, CysH, SirA). Three siderophore BGCs (clusters 12, 19, 22) were annotated, with cluster 12 showing 83% similarity to desferrioxamine B/E clusters. Heavy metal resistance genes were also predicted in silico, suggesting environmental adaptability.

In contrast, the functional and ecological roles of these genetic elements remain largely hypothetical or require future validation:

Plant growth promotion mechanisms (nitrogen fixation, P/K solubilization, H₂S signaling, siderophore-mediated iron uptake) and biocontrol efficacy are predicted in silico but not yet verified in planta or in field settings.

Metabolite activity: While BGCs for streptothricin, candicidin, albaflavenone, and lipopeptides were identified in silico, these compounds have not been isolated from FE-3-1, and their bioactivity is inferred from literature.

Biocontrol applicability:The strain’s proposed broad-spectrum pathogen inhibition and commercial potential (analogous to Actinovate® or Mycostop®) are hypotheses requiring in planta trials, formulation studies, and field efficacy data.

Environmental remediation: Heavy metal immobilization mechanisms are proposed based on genomic annotation but lack experimental confirmation.

Thus, future work must prioritize compound isolation, biochemical characterization, in planta efficacy testing, and field validation to translate genomic potential into practical applications.

Q-5: Reinforce the conclusion with perspective on how this genome resource could contribute to the mining of new actinobacterial biocontrol strains.

R-5: The conclusion section was revised and reorganized to emphasize the potential of this genomic resource in facilitating the discovery of novel actinomycetes-based biological control strains.

The following contents have been added on Page 30-31 Line 539-554, and the changes have been highlighted in red.

This study isolated a Streptomyces strain, FE-3-1, from the rice rhizosphere, demonstrating significant inhibitory activity against the plant pathogen Pyricularia oryzae. Morphological and phylogenetic analysis identified the strain as Streptomyces rochei. Whole genome sequencing revealed a substantial biosynthetic potential, anchored by 31 biosynthetic gene clusters including at least four associated with known antimicrobials. This high-quality genome sequence provides a crucial resource for comparative actinobacterial genomics. Genome mining confirmed the strain's strong capacity for secondary metabolite production, highlighting its value as a source of bioactive compounds. Collectively, these findings provide a robust theoretical foundation for developing Streptomyces rochei FE-3-1 into effective biofertilizers or biopesticides. Furthermore, the genomic insights facilitate the targeted mining of related, underexplored actinobacteria, paving the way for discovering next-generation biocontrol strains with novel or enhanced modes of action against diverse plant pathogens.

---

## [Editor Report · Decision Letter 4]

15 Aug 2025

Isolation, identification, and whole-genome sequencing of Streptomyces rochei FE-3-1 against Pyricularia oryzae

PONE-D-24-32365R4

Dear Dr. Shan,

We’re pleased to inform you that your manuscript has been judged scientifically suitable for publication and will be formally accepted for publication once it meets all outstanding technical requirements.

Kind Regards,

Chetan Keswani, Ph.D.

Academic Editor

PLOS ONE

Additional Editor Comments (optional):

NIL
---

## [Editor Report · Acceptance letter]

PONE-D-24-32365R4

PLOS ONE

Dear Dr. Shan,

I'm pleased to inform you that your manuscript has been deemed suitable for publication in PLOS ONE. Congratulations! Your manuscript is now being handed over to our production team.

Kind regards,

on behalf of

Dr. Chetan Keswani

Academic Editor

PLOS ONE